# A Highly Reliable Convolutional Neural Network Based Soft Tissue Sarcoma Metastasis Detection from Chest X-ray Images: A Retrospective Cohort Study

**DOI:** 10.3390/cancers13194961

**Published:** 2021-10-01

**Authors:** Christoph Wallner, Mansoor Alam, Marius Drysch, Johannes Maximilian Wagner, Alexander Sogorski, Mehran Dadras, Maxi von Glinski, Felix Reinkemeier, Mustafa Becerikli, Christoph Heute, Volkmar Nicolas, Marcus Lehnhardt, Björn Behr

**Affiliations:** 1Department of Plastic Surgery, BG University Hospital Bergmannsheil, Ruhr University Bochum, Bürkle-de-la-Camp Platz 1, 44789 Bochum, Germany; marius.drysch@bergmannsheil.de (M.D.); johannes.wagner@bergmannsheil.de (J.M.W.); alexander.sogorski@rub.de (A.S.); mehran.dadras@rub.de (M.D.); Maxi.vonGlinski@bergmannsheil.de (M.v.G.); Felix.Reinkemeier@ruhr-uni-bochum.de (F.R.); Mustafa.Becerikli@ruhr-uni-bochum.de (M.B.); marcus.lehnhardt@rub.de (M.L.); bjorn.behr@rub.de (B.B.); 2National Centre of Robotics and Automation, University of Engineering & Technology, Peshawar 25000, Pakistan; 12pwmct0216@uetpeshawar.edu.pk; 3Institute for Radiological Diagnostics, Interventional Radiology and Nuclear Medicine, BG University Hospital Bergmannsheil, Ruhr University Bochum, Bürkle-de-la-Camp Platz 1, 44789 Bochum, Germany; christoph.heute@bergmannsheil.de (C.H.); volkmar.nicolas@bergmannsheil.de (V.N.)

**Keywords:** KI, machine learning, python, sarcoma, chest, X-ray

## Abstract

**Simple Summary:**

Soft tissue sarcomas are relatively rare malignant diseases. Part of the diagnosis and follow-up includes medical imaging of the thorax for detection of lung metastases. A Python script was created and trained using a set of lung X-rays and concordant CT scans from a high-volume German-speaking sarcoma center. It is capable of detecting malignant metastasis in the lung with a precision of 71.2%, specificity of 90.5%, sensitivity of 94% and accuracy of 91.2%. Furthermore, the program was able to detect even small nodules with a size <1 cm in conventional X-rays of the thorax. This algorithm was implemented into our daily clinical practice alongside with the radiologists’ findings. With this tool we aim to improve the quality of our service and reduce the expenditure of time.

**Abstract:**

Introduction: soft tissue sarcomas are a subset of malignant tumors that are relatively rare and make up 1% of all malignant tumors in adulthood. Due to the rarity of these tumors, there are significant differences in quality in the diagnosis and treatment of these tumors. One paramount aspect is the diagnosis of hematogenous metastases in the lungs. Guidelines recommend routine lung imaging by means of X-rays. With the ever advancing AI-based diagnostic support, there has so far been no implementation for sarcomas. The aim of the study was to utilize AI to obtain analyzes regarding metastasis on lung X-rays in the most possible sensitive and specific manner in sarcoma patients. Methods: a Python script was created and trained using a set of lung X-rays with sarcoma metastases from a high-volume German-speaking sarcoma center. 26 patients with lung metastasis were included. For all patients chest X-ray with corresponding lung CT scans, and histological biopsies were available. The number of trainable images were expanded to 600. In order to evaluate the biological sensitivity and specificity, the script was tested on lung X-rays with a lung CT as control. Results: in this study we present a new type of convolutional neural network-based system with a precision of 71.2%, specificity of 90.5%, sensitivity of 94%, recall of 94% and accuracy of 91.2%. A good detection of even small findings was determined. Discussion: the created script establishes the option to check lung X-rays for metastases at a safe level, especially given this rare tumor entity.

## 1. Introduction

The term soft tissue sarcoma (STS) comprises a heterogenous group of malignant tumors deriving from a mesenchymal origin. They represent 1% of all malignant tumors in adults [1,2]. The incidence ranges between 1.8 and 5 per 100,000 people. Regional differences in incidence rates are based, among other things, on differences in the ethnic composition of the population. Rhabdomyosarcomas and synovial sarcomas account for around 30% of the soft tissue sarcoma in young adults [3]. The most common diagnoses in older patients are undifferentiated sarcomas, leiomyosarcomas, liposarcomas, fibrosarcomas and pleomorphic sarcomas [4]. Based on the histological subtype, the risk of metastasis and the routes it takes are also determined [5].

Surgical excision is one of the main pillars of therapy for these tumors [6,7]. Nonetheless, a multimodal transdisciplinary approach is important in the treatment of this rare but aggressive form of tumor. Radiation therapists, oncologists, oncopsychologists and radiologists play an important role in the treatment of sarcomas.

The prognosis of a sarcoma depends not only on the entity but also on the extent of the disease. While the 5-year survival probability for a locally limited sarcoma is 81%, it dramatically decreases to 15% for metastatic disease [8,9]. This significant deterioration might also be caused by late detection of filiae. Most commonly sarcomas metastasize to the lungs by means of hematogenous spreading [10]. Routinely, follow-up visits with contrast-enhanced MRI of the local region of interest and posterior-anterior X-ray of the chest is carried out [11]. A CT-scan of the chest is only performed in case of X-ray abnormalities at our institution as recommended by current guidelines [12]. 

Follow-up care and evaluation of the imaging findings is still performed by doctors with varying levels of training and experience in diagnosing this rare tumor. In most clinical settings, the radiologist evaluates the screening X-ray in different cancer entities. However the sensitivity and specificity of radiologists highly varies in chest X-ray. It is important to point out the radiologic difficulties of detection and interpretation of X-ray findings which can actually harm patients [13]. Experienced radiologists might miss up to 10% to 20% of small nodules [14]. Even in CT scans sensitivity and specificity of radiologists do not reach 90% for detecting lung nodules [15].

In recent years, radiological diagnostics have been expanded and quality assured in numerous tumor entities utilizing AI, machine learning and deep learning. These applications display better detection rates than a radiological-medical assessment alone [16]. Modern deep learning frameworks are capable to reach 90% to 95% sensitivity and specificity for nodule detection in CT scans of the lung and around 85% to 90% in chest X-ray [17]. So far there has been no comparable program for the detection of sarcoma metastases in the lungs as most algorithm are focused on lung cancer. Our aim was to establish an algorithm to reliably detect suspicious lung nodules in sarcoma patients based on posterior-anterior chest X-rays and implementing an AI-supported follow-up process.

## 2. Methods

### 2.1. Acquisition of Patient X-ray

Sarcoma patient chest X-ray images were collected retrospectively in accordance with the ethical committee of the Ruhr University Bochum between January 2012 and December 2020. In this period 1129 sarcoma patients were included in our follow up program. Of 110 patients with a regional or distant metastasis of a soft tissue sarcoma, 56 had histologically verified metastasis in the lung, 26 of them had consistent CT-scans where an X-ray was also available. Among these 26 patients, 43 corresponding CT-scans and 43 chest X-rays with sarcoma metastasis nodules were available. The data and patient inclusion as well as the training algorithm is depicted in Figure 1.

### 2.2. Implementation in Python

For this project the “ChestX-ray nodule detection”-dataset was provided from Kaggle (Google LLC., San Francisco, CA, USA) under the CC0 1.0 Universal (CC0 1.0) Public Domain Dedication in order to detect lung nodules in X-rays. Total images of X-rays were 1500 which we then split for training and evaluation. 1400 X-rays with additional 43 sarcoma nodule positive images from our clinic were used for training and 100 images of X-rays were used for evaluation. The number of in-house training images was artificially increased from 43 to 600 by randomized partial morphing. This method to generate synthesized data was described before [18]. The algorithm used for detection is YOLOv5s, which is a compound-scaled object detection model trained on COCO datasets and includes model assembling and hyperparameter evolution [19,20]. It comprises the following three main parts: Module BackBone is used to extract features from the images that are input. The module Neck is used for making pyramid features which is used for generalization, while the Head is used for the detection process (see Figure 2).

During the training of YOLOv5s for nodule detection, total numbers of epochs were 100 and batch sizes were 2. For all programming tasks, Python 3.8 with pytorch library was used.

### 2.3. Validation of the Model 

In order to validate our model, 50 lung X-rays with underlying CT images were used as gold standard with high-grade suspicious foci which were subsequently verified by histological diagnosis, including 15 images with sarcoma metastases in the lung verified by both a CT scan (Somaton Defintion Edge, Siemens Healthineers, Erlangen, Germany) and concordant chest X-ray. These images were not included into our training model and served as external control. Specificity, precision and recall were further evaluated by analysis of 200 lung X-rays with underlying CT images without radiological evidence of metastasis as the gold standard. The indication for the CT scans was made in the course of other diseases such as lung cancer, accidents or during an intensive care stay without the radiological validation of a suspicious mass. The CT-scans were performed with patient in supine position and inspirational hold (slice thickness = 1 mm, tube voltage = 100 kVp).

### 2.4. Data Managment

Data management was performed using GraphPad PRISM (version: 8.3.0; Graphpad Software, Inc., La Jolla, CA, USA).

## 3. Results

### 3.1. Patient Data

To test the trained algorithm, 26 patients with lung metastasis of sarcoma and according CT scans and chest X-rays obtained in a close time interval (<2 weeks) were included. Those 26 patients had 43 CT scans and 43 corresponding chest X-rays. Patient characteristics are presented in Table 1. The median age was 62.8 years, while 58% of the patients were male. Half of all tumor entities were undifferentiated pleomorphic sarcoma. 74% of all tumors in the collective were G3.

### 3.2. Evaluation of Sensitivy, Specificity, Precision and Accuracy

Biological sensitivity and specificity were determined by utilizing CT images which were obtained as part of the follow-up care for chest X-rays suspicious for distant metastasis. Negative controls were obtained from non-tumor-associated CT images related to trauma or non-malignant lung disease as given in on daily base at an emergency hospital. These images were not included into our training model and serving as external validation. All nodules detected by the application with a test batch prediction over 0.5 were included. Results of the sensitivity and specificity tests are presented in Figure 3. F1 score ROC (Receiver Operating Characteristic) and accuracy were utilized as evaluation metrics. This is shown in Figure 4. Sensitivity was shown with 94%, specificity with 90.5%, precision with 71%, recall with 94% and accuracy with 91%.

### 3.3. Smallest Reliably Detectable Size

In order to check the validity of the algorithm, X-rays of the lungs in the course of the follow-up care were compared with the CT examinations which were performed in the event of suspicious nodules. For this purpose, findings were assigned to the size and location in the CT. Smaller findings, which the radiologist did not explicitly describe in the X-ray but were described in the CT, served as a size reference for the sensitivity measurement. All findings detected by the algorithm in X-rays were compared with those found in CTs but previously overlooked in the X-ray. In this way, a sensitivity for the various findings sizes was established (see Figure 5). 

### 3.4. Implementation of a Hybrid Follow-Up Algorithm

The follow-up of our soft tissue sarcoma patients is based on the UK and ESMO-guidelines [21]: After initial treatment patients receive a follow up with a contrast-enhanced MRI of the local region and a lung X-ray every 3 months in the first 2 years, then twice a year up to the fifth year, and once a year thereafter when necessary [22]. This is also stated in the ESMO (European Society for Medical Oncology) and German guidelines for the treatment of soft tissue sarcoma [22,23].

We identified the evaluation of chest X-ray by a plastic surgeon and a radiologist in the daily routine as a potential vulnerability in the quality of our follow-up. Therefore, we implemented a new hybrid process to strengthen the quality of our follow-ups. Figure 6 shows the implementation of the AI-framework into our routine follow-up of sarcoma patients. In this setting the patient’s chest X-ray is evaluated by a radiologist and simultaneously by the AI-framework. The results of both processes are validated and interpretated by the plastic surgeon who does the consultation of the patients follow-up. 

## 4. Discussion

As one of the high-volume sarcoma centers in German-speaking countries, it was possible to train and verify a new algorithm for sarcoma lung metastasis detection with a corresponding number of recordings. The reliability in the detection of smaller tumors proved particularly good. However, there was a difficulty in detecting exceptionally large nodules and masses. These could be confused with mediastinal structures. We acknowledge the poorer sensitivity in large masses using the training model, which was designed for metastases. This is the reason why tumors larger than 5 cm had a worse detection rate than tumors between 1 and 5 cm in this study. Large masses can also be misinterpreted as, for example, atelectasis [6]. Nevertheless, there is no fundamental approach-based problem here, as a sensitivity of 83% in the large tumor masses can also be explained by the small number of cases of these large tumor foci (*n* = 16) in the study. 

On the other hand ROC-curve, as well Precision-Recall curve of the model showed a weakness due to the imbalance of a positive and negative samples in the training. This can be further improved by including more positive samples with cooperation of more centers including more positive cases. Another possibility to reduce the imbalance would be random undersampling of the nodule free images. We aimed to avoid this due to the chance that potential useful information will be deleted.

One major limitation of this study is the small number of sarcoma metastasis positive CT-scans verified by histology. Even though being one of the largest sarcoma centers in Germany, a cooperation with other specialized clinics may improve the algorithm and therefore sensitivity and specificity. With large databases such as Kaggle, the international community can profoundly benefit in terms of a good exchange of datasets. AI in particular needs as much training data as possible in order to deliver valid results. With increasing amounts of data, but also with the ever ameliorated AI, hardware is being used to its capacity. More complex algorithms require powerful graphics processing unit (GPU). The advantage of this algorithm described is the resource-saving application with a computing time of less than 1 s with nevertheless reliable results.

The method of convolutional neural networks (CNN) developed a few years ago has revolutionized AI and particularly image recognition and significantly improves medical applications. CNNs are inspired by biological processes (i.e., interactive activity of neurons in an animal brain). In brief, they consist of an input layer, in-between hidden layers and an output layer. The convolution (process of computing in CNNs) takes place in the hidden layers [24]. Especially in the context of the COVID-19 pandemic, these supervised machine learning applications have been significantly expanded and advanced [25]. On the one hand, CNN leads to a significantly faster computing time, but also to a higher sensitivity and specificity. In the recent literature microscopic brain tumor detection was achieved with a 3d CNN and feature selection architecture [26].

With the sensitivity of 94% and a specificity of 90.5% on posterior-anterior chest X-rays presented in our project, values comparable to CT-scans can be achieved [27]. In addition, through the training of the sarcoma-based X-rays we provided, the technical sensitivity and specificity in the training images was increased by 19.5% compared to test runs without the 600 images implemented. This would not have been so sensitive and specific without the additional 600 images added to the online data-set of 1400 images. If nodules with a test batch prediction of less than 0.5 had been included, the sensitivity would have increased at the expense of the specificity.

Large sarcoma centers typically have experienced radiologists who can also detect small findings on chest X-rays. A study by Gamboa et al. demonstrated no improvement in the survival outcome from a CT scan instead of an X-ray in the follow-up [28]. This underlines the importance of X-rays in the context of diagnostics—with the requirement of an experienced radiologist. However, the software developed enables the comprehensive detection of metastases in the X-ray without this experience. The rate of overlooked findings is also reduced using AI to detect cancer metastatis [29]. For resident radiologists, in particular those with little experience, an expansion of diagnostics using AI could generate reliability and security. Despite sophistication of modern machine learning applications in medicine, especially medical imaging, the contextual and responsive competence of the radiologist is still indispensable.

Another important aspect of sarcoma therapy and follow-ups is properly assessing the metastatic potential of various entities and de-differentiation. For the risk of a distant metastasis tumor size, high histologic grade, deep location, recurrent disease at presentation, the subtype of leiomyosarcoma and non-liposarcoma histology were independent adverse prognostic factors [30]. While other entities such as dermatofibrosarcoma protuberans rarely metastasize [31]. Based on our data half of all patients with pulmonal metastasis had the diagnosis of a pleomorphic sarcoma, an advanced age and higher grading. The AI-framework in this project was only trained based on the radiological material given. The efficiency of the AI can be further improved by including prognostic factors. At this point, a link between artificial intelligence and big data can be established. Big data was for example used in the planning of cancer immunotherapy [32]. Distillation of data given in the sarcoma registers can improve AI based cancer therapy. In this context, networking of the sarcoma centers for this rare disease in order to generate big data resources should be promoted.

With integration of the hybrid follow-up process into the clinical routine the follow-up procedure was theoretically improved. In the literature the improvement in diagnostics through AI in lung cancer could be shown [33,34]. Based on the numbers in the literature, an improvement in sensitivity and specificity of around 5% can be assumed. In our scenario with around 800 sarcoma follow ups per year an improvement of 5% bears a game-changing effect [14,34]. A subsequent study is needed to evaluate the eventual effect of this hybrid follow-up process. At this point, it is possible to guarantee at least a more structured sequence in everyday clinical practice for up to 10 sarcoma patient follow ups per day.

Further applications of the CNN can be considered in terms of classification and segmentation of sarcoma including multiple layers of information [35,36]. We aim to improve our follow-up consultations by evaluating the MRI of the local region with an AI-framework as established with other tumor entities [37,38]. Zhang recently developed a CNN based program for sarcoma grading and metastasis detection in the lungs [39]. Several steps of sarcoma diagnosis and follow-ups can be guided by AI. Embracing the whole diagnosis and therapy pathway, AI can be implemented into literature or guideline implementation, analysis of the histology, setting up the therapy plan, radiological evaluation and AI-supported patient education. There is a cornucopia of innumerable possibilities to support the cancer therapy through integration of AI.

With regards to the literature including our presented results we do not see any advantage of the CT over the conventional chest X-ray within the regular follow-ups. Sensitivity and Specificity of the X-ray reach levels of the CT scan with considerable less exposure to radiation and significant resource savings [28].

## 5. Conclusions

In the recent years the research on artificial intelligence driven diagnostic and prognostic tools has exponentially grown. To date there are no established applications for sarcoma follow-ups known. With inclusion of the CNN assisted evaluation of chest X-ray the follow-ups can be improved. Further research is required to show systemic benefits of this hybrid follow-up.

## Figures and Tables

**Figure 1 cancers-13-04961-f001:**
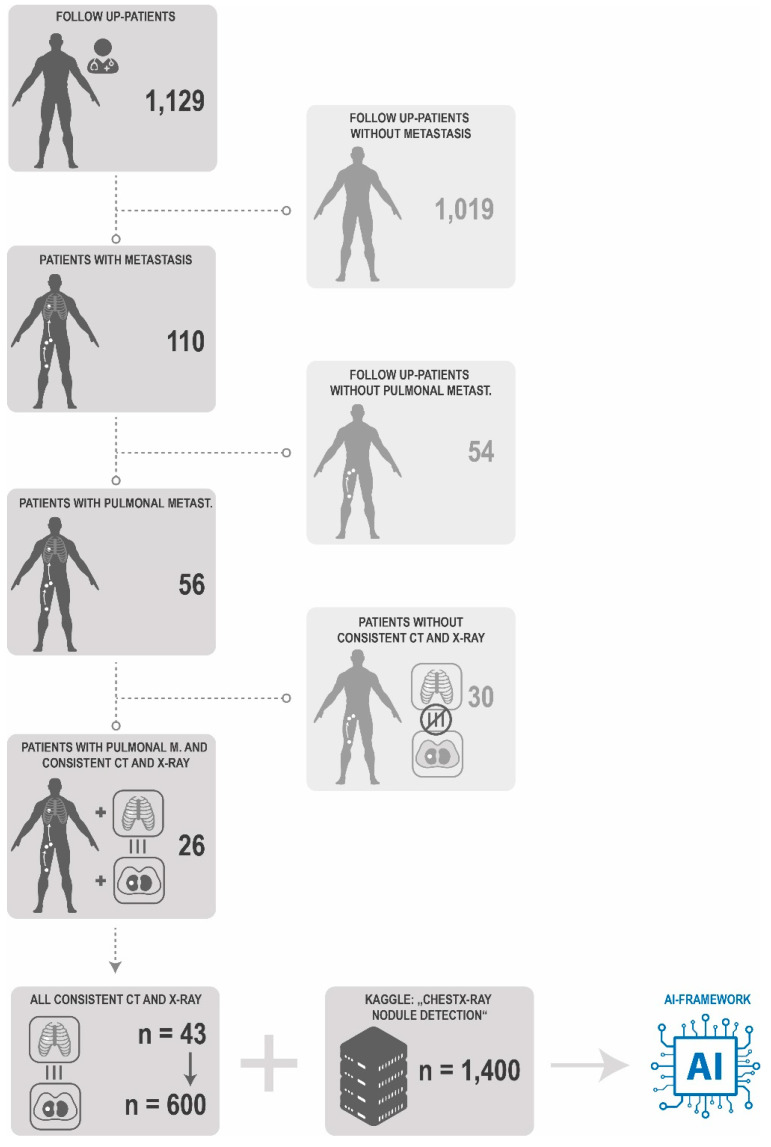
Patient and Data Inclusion. In total 1129 patients underwent follow-up in our clinics. 1019 were detected without any metastasis over the period 2012–2020. 110 patients had radiological signs for a sarcoma metastasis. 54 patients of them without pulmonal metastasis were excluded. 56 patients had pulmonal metastasis, while only 26 patients had consistent and temporally corresponding CT scans and X-rays. 30 patients of the 56 patients with pulmonal metastasis had either no CT scan or an x-ray or weren’t temporally corresponding. Of the 26 patients with histologically verified pulmonal metastasis 43 pairs of corresponding CT scans and x-ray (multiplication up to 600 through transformation) showing a sarcoma metastasis were included into the learning process of the AI framework together with the weighted Kaggle “ChestX-ray nodule detection”-dataset.

**Figure 2 cancers-13-04961-f002:**
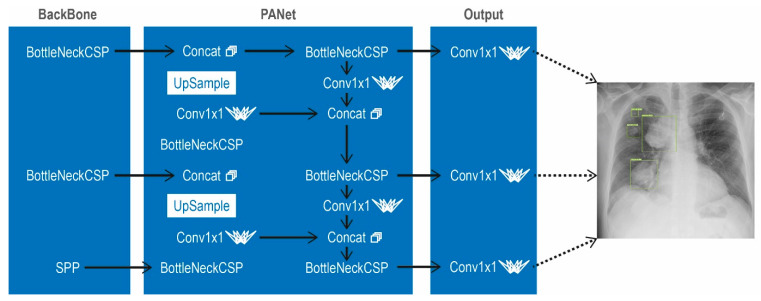
Scheme of the YOLOv5 Architecture as Convolutional Neural Network (CNN). Main parts include the BackBone, Neck and Head. In the BackBone, CSPNet is used in order to extract features from the images which are used as input images. The Neck is used for the creation of pyramid feature. It helps the module on scaling factor of detected objects which are of the same nature but different scales. The technique which is used for creation of pyramid features is PANet. The main function of the head module is to apply anchor of different sizes on those features which are generated in the previous layers with value of probability as well as bounding box with score. SPP stands for Spatial Pyramid Pooling, Concat stands for concatenation.

**Figure 3 cancers-13-04961-f003:**
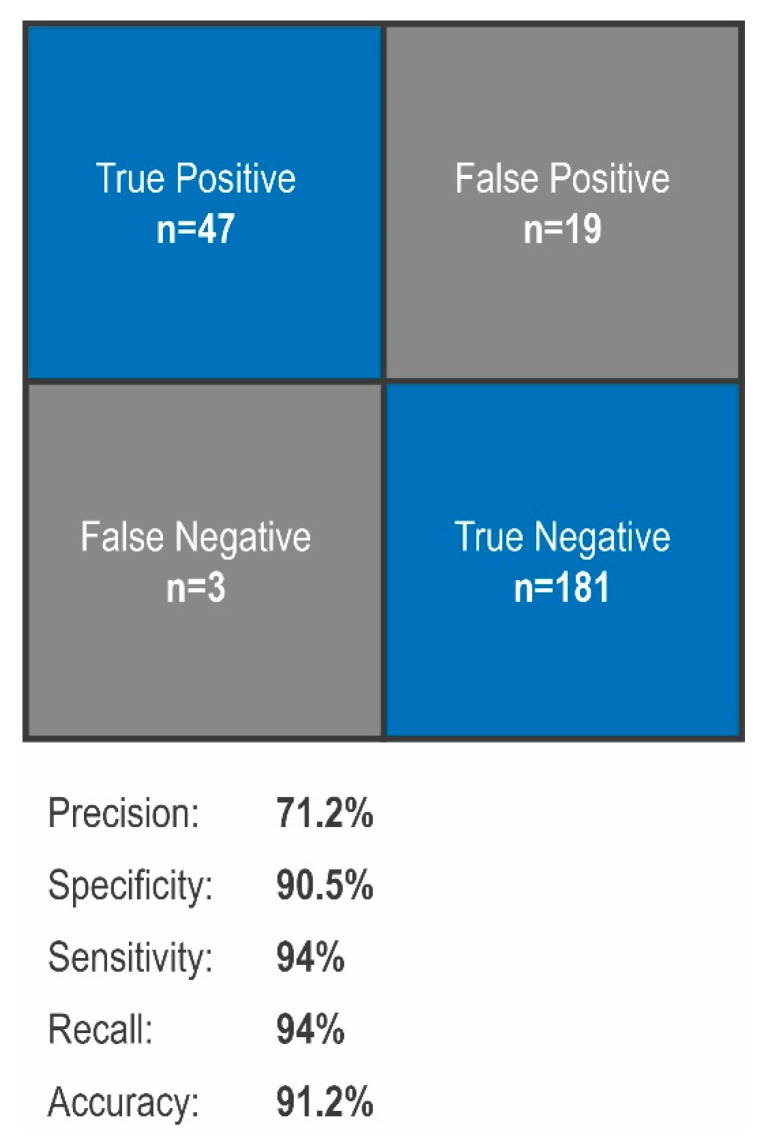
Validation of the Algorithm. Utilizing 200 nodule-free images and 50 images positive for nodules for external validation, precision of 71.2%, specificity of 90.5%, sensitivity of 94%, recall of 94% and accuracy of 91.2% could be determined.

**Figure 4 cancers-13-04961-f004:**

ROC Curve, Precision-Recall Curve, F1-Score. The ROC curve of the model shows an area under the curve of. The ROC-Curve shows and area und the curve (AUC) of 0.8761, the Precision-Recall Curve shows an AUC of 0.7852, while the threshold of 0.5 shows a F1-score of 0.7321.

**Figure 5 cancers-13-04961-f005:**
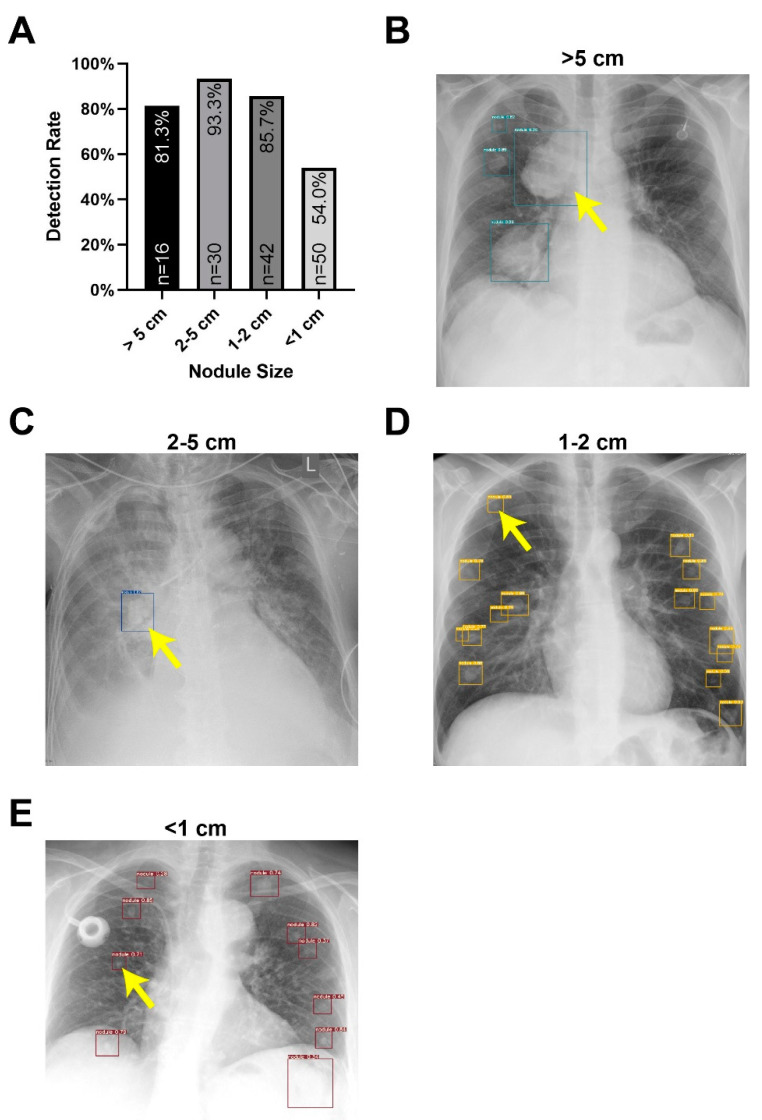
Smallest Reliably Detectable Size. (**A**) Findings in the CT were assigned to different groups according to size. Based on the X-ray, the algorithm was intended detect all findings. The sensitivity was divided into the groups according to the findings found and overlooked. While the program reliably detected nodules between 1 and 5 cm in over 93.3% and 85.7%, respectively, nodules larger than 5 cm and especially smaller than 1 cm were more challenging for the algorithm. (**B**) Lung X-ray of a finding over 5 cm. (**C**) Lung X-ray of a finding between 2–5 cm. (**D**) Lung X-ray of a finding between 1–2 cm. (**E**) Lung X-ray of a finding under 1 cm. Yellow arrows mark the findings.

**Figure 6 cancers-13-04961-f006:**
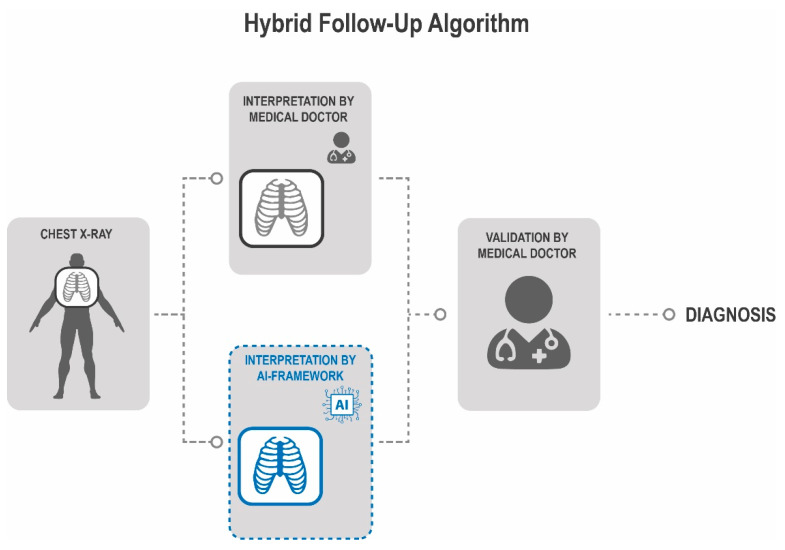
Hybrid Follow-Up Algorithm. In order to improve our follow-up, the AI-based evaluation of the X-ray is placed parallel to the radiological assessment to avoid any bias. The information from both interpretations are validated by an independent second medical doctor.

**Table 1 cancers-13-04961-t001:** Study population of patients with lung metastasis of a soft tissue sarcoma with concordant CT-scan and Chest X-ray (*n* = 26).

	Value
Age at X-ray (years)	62.8 ± 12.2
Sex	
Male	15 (58%)
Female	11 (42%)
Sarcoma Entity	
Undiff. Pleomorph. Sarcoma	13 (50%)
Liposarcoma	2 (8%)
Myoxid Sarcoma	3 (12%)
Leiomyosarcoma	3 (12%)
Malignant fibrous histiocytoma	4 (15%)
Other Sarcoma Entitiy	1 (3%)
Tumor Grade	
G1	2 (8%)
G2	5 (18%)
G3	19 (74%)

## Data Availability

The datasets used and/or analyzed during the current study are available from the corresponding author on reasonable request.

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
