# Peer review of "A Highly Reliable Convolutional Neural Network Based Soft Tissue Sarcoma Metastasis Detection from Chest X-ray Images: A Retrospective Cohort Study"

_cancers, 2021, doi:10.3390/cancers13194961_

Round 1
Reviewer 1 Report
1) The authors investigate CNN detection of sarcoma metastases on chest X ray images. This is interesting because previous studies in sarcoma image data science have focused on volumetric studies such as MRI, CT, and PET and generally involve representation learning of the primary tumors, not lung scans for metastasis detection. The proposed research could be useful for following sarcoma patients following their treatments.
A limitation is that chest X ray surveillance could be supplanted by low dose CT surveillance in this population in the future. Further, other sarcoma studies are larger and some have external validation. While these studies tended to focus on empirical image features instead of data driven approaches such as CNN, empirical image studies may be practical versus high dimensional machine learning architectures in small patient cohorts.
2) The authors evaluate 26 patients with lung metastasis. The dataset is too small to transfer this architecture. Some patients have multiple scans and so the authors evaluate 43 image sets. No external validation is used.
The data are highly skewed. The authors have 1400 scans belonging to the majority class (no mets) with 26 patients and 43 unique images belonging to the minority class (mets). The authors use data augmentation to generate 600 images in the minority class. In particular the failures in detection in nodules with size >5cm indicates that enlargement may not have been an appropriate augmentation strategy. Likely their high prediction rates are a function of this high augmentation rate.
The authors do not describe their validation work in sufficient depth. The work would be improved by larger studies, held out datasets, and external validation datasets.
3) The performance of the X ray model alone does not appear to be reported. Instead the authors appear to report the “biological sensitivity” results from 2.3 using CT imaging as a gold standard (Fig 3). The descriptions are not adequate here and it is difficult because many or all figure references are broken: "Error! Reference source not found" or missing. The authors should carefully review the description of the construction of their models and reporting of results.
4) The literature discussion needs substantial revision. The authors should address the existing literature in sarcoma image evaluation to predict prognosis, tumor grade, metastatic progression, etc. While these studies generally to look at the primary tumor instead of lung screening, they are sufficiently related to the subject at hand. A few examples:
-The authors should reference previous reports related to image prediction of sarcoma metastasis using empirical image features such as Vallieres Phys Med Biol. 2015 Jul 21;60(14):5471-96. doi: 10.1088/0031-9155/60/14/5471. Epub 2015 Jun 29.
-There are previous reports of applying CNN to sarcoma MR imaging for different applications such as Navarro Cancers (Basel). 2021 Jun 8;13(12):2866. doi: 10.3390/cancers13122866.
-It is unclear how refs 35-37 apply. Consider other reports in sarcoma image data science.
5) The written text is generally acceptable. There are some expressions that don’t come across well such as “security loophole” in section 3.4. There are other questions on reporting methods and results raised elsewhere.
Author Response
- 1) The authors investigate CNN detection of sarcoma metastases on chest X ray images. This is interesting because previous studies in sarcoma image data science have focused on volumetric studies such as MRI, CT, and PET and generally involve representation learning of the primary tumors, not lung scans for metastasis detection. The proposed research could be useful for following sarcoma patients following their treatments.
A limitation is that chest X ray surveillance could be supplanted by low dose CT surveillance in this population in the future. Further, other sarcoma studies are larger and some have external validation. While these studies tended to focus on empirical image features instead of data driven approaches such as CNN, empirical image studies may be practical versus high dimensional machine learning architectures in small patient cohorts.
Our answer:
Dear Reviewer,
we thank the reviewer for summarizing up our study. We have implemented the low dose CT scans as control for the x-ray in a temporal proximity serving as an appropriate validation. This has been described before and recently in a large-scale validation for the diagnostic accuracy of Artificial Intelligence-Assisted CT imaging in COVID-19 disease [1]. Furthermore, we have implemented an external evaluation as described under point 2.
2) The authors evaluate 26 patients with lung metastasis. The dataset is too small to transfer this architecture. Some patients have multiple scans and so the authors evaluate 43 image sets. No external validation is used.
The data are highly skewed. The authors have 1400 scans belonging to the majority class (no mets) with 26 patients and 43 unique images belonging to the minority class (mets). The authors use data augmentation to generate 600 images in the minority class. In particular the failures in detection in nodules with size >5cm indicates that enlargement may not have been an appropriate augmentation strategy. Likely their high prediction rates are a function of this high augmentation rate.
The authors do not describe their validation work in sufficient depth. The work would be improved by larger studies, held out datasets, and external validation datasets.
Dear Reviewer,
we thank for the comment on our validation. Through including the open source (CC0: Public Domain) Kaggle “ChestX-ray nodule detection”-dataset we were able to upscale the algorithms sensitivity and specificity to detect nodules. However, to strengthen the weight of our images they were enlarged to 600 synthesized images by randomized partial morphing. This information is included in the paragraph “Implementation in Python” in the manuscript. We added more information to this paragraph in the updated manuscript to legitimate this method. This method has been described before and is particularly useful in rare clinical pictures and therefore small data sets [2,3]. Thus, despite being one of the largest German sarcoma centers and the long observation period, it is only possible to generate a large data set with pulmonary metastatic X-rays and correlating CT-scans to a limited extent.
Regarding the validation, we apologize for the confusion we have caused and improved our description about the external validation we performed. We included a more detailed description about the external validation in the new paragraph: Validation of the model. We clarified that 50 images with detected metastasis in the lung through CT-scans and X-ray were included into the external validation. These images were not used in the training model. 15 of those images included sarcoma metastasis. For defining the specificity, we included X-rays of patients with nodule-free CT-Scans, which were not included into the training. As recently published, studies are using radiological reports as reference standard for external validation, which we would not rely on [4]. In a meta-analysis by Zheng et al. of only 7 of 34 studies for metastasis detection in chest X-ray used an external validation [5]. Given those points, we think we now included a sufficient internal and external validation into our manuscript.
We see the poorer sensitivity in large masses using the training model, which was designed for metastases. Large masses can also be misinterpreted as, for example, atelectasis [6]. Nevertheless, there is no fundamental approach-based problem here, as a sensitivity of 83% in the large tumor masses can also be explained by the small number of cases of these large tumor foci in the study. We included this statement into the discussion of the revised manuscript. Reviewer 2 recommended to include ROC curve, F1-score and Precision-Recall-Curve. These provide more information as seen in the responses to reviewer 2.
3) The performance of the X ray model alone does not appear to be reported. Instead the authors appear to report the “biological sensitivity” results from 2.3 using CT imaging as a gold standard (Fig 3). The descriptions are not adequate here and it is difficult because many or all figure references are broken: "Error! Reference source not found" or missing. The authors should carefully review the description of the construction of their models and reporting of results.
Dear Reviewer,
we apologize for the confusion we have caused. With the validation we use through a more sensitive method (CT scan) than the target method (chest X-ray), we can deliver a higher level of security than through a human input or out of sample images. Furthermore, our reference standard is not only based on one feature – as most other studies - but three: histopathology, CT-scan and expert consensus [5]. We also included as recommended by reviewer 2 ROC curve, F1-score and Precision-Recall-Curve.
The images are meant to clarify the process. We double checked them so that they are visible.
4) The literature discussion needs substantial revision. The authors should address the existing literature in sarcoma image evaluation to predict prognosis, tumor grade, metastatic progression, etc. While these studies generally to look at the primary tumor instead of lung screening, they are sufficiently related to the subject at hand. A few examples:
-The authors should reference previous reports related to image prediction of sarcoma metastasis using empirical image features such as Vallieres Phys Med Biol. 2015 Jul 21;60(14):5471-96. doi: 10.1088/0031-9155/60/14/5471. Epub 2015 Jun 29.
-There are previous reports of applying CNN to sarcoma MR imaging for different applications such as Navarro Cancers (Basel). 2021 Jun 8;13(12):2866. doi: 10.3390/cancers13122866.
-It is unclear how refs 35-37 apply. Consider other reports in sarcoma image data science.
Dear Reviewer,
we thank for the suggestions to enrich our references. However, the references you recommended Vallieres et al. as well as Navarro et al. only describe the application of a CNN in MRI images of a original tumor site. The separation between local staging (MRI of the local findings) and system staging (X-ray of the lungs) must be clarified at this point [7]. Our model is designed solely for the detection of tumor metastases in the lungs using X-rays.
We're talking about two completely different technical and diagnostic approaches. The only commonality between these studies and our study is the inclusion of AI and the entity of sarcoma. However, we included the reference by Navarro et al. as he also used multiple layers of a CNN as we did.
We included references 35-37 of the original manuscript as they describe the application of CNN in in radiography especially of the lung. We follow the advice of the reviewer and have removed reference 37, as only the application of the multilayer CNN to brain tumors was shown here. Again, we want to clarify the focus of our work on the sarcoma metastasis in the lungs and not on the entity of the sarcoma in different regions, which is further diagnosed using MRI.
5) The written text is generally acceptable. There are some expressions that don’t come across well such as “security loophole” in section 3.4. There are other questions on reporting methods and results raised elsewhere.
We thank the reviewer for this suggestion and changed this expression in paragraph 3.4 of the revised manuscript
Reviewer 2 Report
The authors presented the results of a convolutional neural network model for predicting pulmonary metastasis in adult patients with soft tissue sarcomas. They have reported a sensitivity and specificity of over 90% in a validation dataset. The advantage of the model is that it increases the accuracy in detecting pulmonary metastasis in selected oncology patients, especially when medical personnel are not adequately trained (e.g., residents, fellows). In addition, a warning (hybrid) model is proposed that generates high suspicion on protocol chest radiographs.
The model is robust and was built using the Python language and libraries, which are free to all and supported by a large community.
However, there are several aspects that need to be corrected:
1) The terminology and metrics used to validate the model are not appropriate for artificial neural networks. The authors should use 'precision' (i.e. PPV=TP/(TP =FP)), 'recall' (i.e. sensitivity, but it is advisable to use the term recall for better comparison), 'accuracy' ((TP +TN)/(TP +TN+ FN =FP)), and 'F1 score' (harmonic mean of precision and recall, which is 80% for this model).
2) Instead of specificity, precision must be reported, which is 71% in this model, and accuracy, which is 91%. The authors should discuss why the precision of their model is so much lower than the recall. This is mainly due to the unbalanced training dataset (the ratio between normal and pathological observations is 1400:600=7:3). In this dataset, positive observations are in the minority and there are many negative observations that could become false positives. However, recall is more important than precision when the cost of acting (chest X-ray) is low but the opportunity cost of passing over a candidate is high (overlooking the metastatic nodule in the lung). The authors should propose undersampling techniques (reducing the number of negative ground truth observations) and then training the classifier.
3) I would recommend showing the ROC and the precision/recall curve (one figure, side by side) for different treshold probabilities (the chosen treshold is 0.5). Also, the F1 curve for different thresholds could help in choosing an appropriate treshold value (when F1 is highest).
4) The authors chose to use a published dataset for the ground truth of normal chest radiographs. It is not mandatory, but it is recommended to publish their dataset of 600 images with metastatic nodes (along with the validation set) on a similar repository so that other researchers can build their models or improve this one . I would also recommend publishing the code on GitHub to improve the reproducibility and quality of their work.
5) It's clear what the training set was: 1400 normal chest X-rays and 600 images from 43 sarcoma patients. But what was the validation data set (paragraph 2.3)? It is not clear. Did the authors have other chest x-rays that served as ground truth observations, or did they use the fraction of the original dataset? And for what purpose did the authors use 100 X rays from the Kaggle dataset "ChestX-ray nodule detection" that were not used in the training set? The first sentence in paragraph 2.3. should start with "In order the validate our model,....".
6) The authors should correct referencing to the tables and figures and replace the error messages with the appropriate text.
Author Response
The authors presented the results of a convolutional neural network model for predicting pulmonary metastasis in adult patients with soft tissue sarcomas. They have reported a sensitivity and specificity of over 90% in a validation dataset. The advantage of the model is that it increases the accuracy in detecting pulmonary metastasis in selected oncology patients, especially when medical personnel are not adequately trained (e.g., residents, fellows). In addition, a warning (hybrid) model is proposed that generates high suspicion on protocol chest radiographs.
The model is robust and was built using the Python language and libraries, which are free to all and supported by a large community.
However, there are several aspects that need to be corrected:
1) The terminology and metrics used to validate the model are not appropriate for artificial neural networks. The authors should use 'precision' (i.e. PPV=TP/(TP =FP)), 'recall' (i.e. sensitivity, but it is advisable to use the term recall for better comparison), 'accuracy' ((TP +TN)/(TP +TN+ FN =FP)), and 'F1 score' (harmonic mean of precision and recall, which is 80% for this model).
Our answer:
Dear Reviewer,
we thank the reviewer for clarifying the terminology and corrected it accordingly in our revised manuscript. We added and illustrated the precision, recall, accuracy and F1 value into our revised manuscript.
2) Instead of specificity, precision must be reported, which is 71% in this model, and accuracy, which is 91%. The authors should discuss why the precision of their model is so much lower than the recall. This is mainly due to the unbalanced training dataset (the ratio between normal and pathological observations is 1400:600=7:3). In this dataset, positive observations are in the minority and there are many negative observations that could become false positives. However, recall is more important than precision when the cost of acting (chest X-ray) is low but the opportunity cost of passing over a candidate is high (overlooking the metastatic nodule in the lung). The authors should propose undersampling techniques (reducing the number of negative ground truth observations) and then training the classifier.
Dear Reviewer,
we thank for this very valuable suggestion and added the fact that our precision is lower due to a smaller positive sample size in the training model. This was added to the discussion of the revised manuscript and can be seen in the ROC and Precision-Recall-Curve. Furthermore, we added the precision, accuracy values to the revised manuscript. However, we would still include sensitivity and specificity into our manuscript as these are valuable information for clinicians.
One simple undersampling technique would involve randomly selecting examples from the nodule free images and deleting them from the training dataset. Although simple and effective, a limitation of this technique is that examples are removed without any concern for how useful or important they might be in determining the decision boundary between the classes. We aimed to avoid this as it is possible, or even likely, that useful information will be deleted. We added this information to discussion of the revised manuscript.
As we describe in section 3.4 we do not rely on our script alone and included this into a hybrid follow up.
3) I would recommend showing the ROC and the precision/recall curve (one figure, side by side) for different treshold probabilities (the chosen treshold is 0.5). Also, the F1 curve for different thresholds could help in choosing an appropriate treshold value (when F1 is highest).
Dear Reviewer,
we are grateful for this recommendation and added the ROC and precision-recall curve to our revised manuscript as proposed. We found the chosen threshold of 0.5 to be one of the highest F1-scores. Therefore, we decided to choose this threshold in the first place.
4) The authors chose to use a published dataset for the ground truth of normal chest radiographs. It is not mandatory, but it is recommended to publish their dataset of 600 images with metastatic nodes (along with the validation set) on a similar repository so that other researchers can build their models or improve this one . I would also recommend publishing the code on GitHub to improve the reproducibility and quality of their work.
Dear reviewer, we thank for this valuable contribution. After acceptance for publication, we will also publish the source code of the python script and the 600 images.
5) It's clear what the training set was: 1400 normal chest X-rays and 600 images from 43 sarcoma patients. But what was the validation data set (paragraph 2.3)? It is not clear. Did the authors have other chest x-rays that served as ground truth observations, or did they use the fraction of the original dataset? And for what purpose did the authors use 100 X rays from the Kaggle dataset "ChestX-ray nodule detection" that were not used in the training set? The first sentence in paragraph 2.3. should start with "In order the validate our model,....".
Dear reviewer, we apologize for the confusion we have caused. Reviewer 1 brought up the same question. We clarified that 50 images with detected metastasis in the lung through CT-scans and X-ray were included into the external validation. These images were not used in the training model. 15 of those images included sarcoma metastasis. For defining the specificity, we included X-rays of patients with nodule-free CT-Scans. We didn’t use and of the Kaggle images for the external validation.
We also changed the first sentence of paragraph 2.3.
6) The authors should correct referencing to the tables and figures and replace the error messages with the appropriate text.
Dear reviewer, we apologize for this technical error and changed it accordingly in the revised manuscript.

Reviewer 3 Report
Dear Editor,
The article " A Highly Reliable Convolutional Neural Network Based Soft Tissue Sarcoma Metastasis Detection from Chest X-ray Images: A retrospective cohort study" presented for review is interesting. The authors covered this issue very well in their article.
I suggest a few corrections:
- Section: 1. Acquisition of Patient X-ray. Was approval from the Bioethics Committee required? If yes, please provide number. No continuity of text: “The data and patient inclusion as well as the training algorithm is depicted in…”
- Section 2.2 Implementation in Python. “…see Error! Reference source not found” please explain.
- Section: 4. Statistical Analysis. What statistical tests the GraphPad PRISM application allows you to perform?
- Section:1. Patient Data. “Patient characteristics are presented in Error! Reference source not found..”, “Results of the sensitivity and specificity tests are presented in Error! Reference source not found..” please explain.
- Section: 3. Smallest Reliably Detectable Size. “In this way, a sensitivity for the various findings sizes was established (see Error! Reference source not found.).” please explain.
- Section: 4. Implementation of a hybrid follow-up algorithm. “Therefore, we implemented a new hybrid process to close this security loophole. Error! Reference source not found.” please explain.“…the implementation of the AI-framework…” please explain.
After corrections, the article may be accepted for publication.
Author Response
The article " A Highly Reliable Convolutional Neural Network Based Soft Tissue Sarcoma Metastasis Detection from Chest X-ray Images: A retrospective cohort study" presented for review is interesting. The authors covered this issue very well in their article.
I suggest a few corrections:
- Section: 1. Acquisition of Patient X-ray. Was approval from the Bioethics Committee required? If yes, please provide number. No continuity of text: “The data and patient inclusion as well as the training algorithm is depicted in…”
Our answer:
Dear Reviewer,
we thank the reviewer for the approval of our work. As stated under “Institutional Review Board Statement” no ethical approval was required.
We changed the syntax as proposed by the reviewer in the revised manuscript.
- Section 2.2 Implementation in Python. “…see Error! Reference source not found” please explain.
Dear reviewer, we apologize for this technical error and changed it in the revised manuscript accordingly.
- Section: 4. Statistical Analysis. What statistical tests the GraphPad PRISM application allows you to perform?
Dear reviewer, we apologize for the confusion we have caused. We excluded the statement about the Chi-Squared-Test as we enriched the information about the algorithm validation by adding recall, precision, and accuracy. However, we have used GraphPad PRISM for illustration. We rephrased section 2.4 to Data management.
- Section:1. Patient Data. “Patient characteristics are presented in Error! Reference source not found..”, “Results of the sensitivity and specificity tests are presented in Error! Reference source not found..” please explain.
- Section: 3. Smallest Reliably Detectable Size. “In this way, a sensitivity for the various findings sizes was established (see Error! Reference source not found.).” please explain.
- Section: 4. Implementation of a hybrid follow-up algorithm. “Therefore, we implemented a new hybrid process to close this security loophole. Error! Reference source not found.” please explain.“…the implementation of the AI-framework…” please explain.
After corrections, the article may be accepted for publication.
Dear reviewer, we apologize for these technical errors and changed them in section 1, 3 and 4 of the revised manuscript.

Round 2
Reviewer 1 Report
Some issues with clarity of text remain which are at the discretion of the editorial office.
Author Response
Dear Reviewer,
we thank the reviewer for the effort to review our updated manuscript. The last manuscript has many changes implemented according to the other reviewers suggestions. These could possibly be contrary to the changes proposed by reviewer 1. However, since these are not listed of a fundamental nature, we assume that the text is accordingly clear. Nonetheless, the text was revised again and checked for spelling errors or leaps in thought and marked as changes. We hope that the updated manuscript is now even clearer and more stringent.
We thank you for considering our work for your journal.
